# Description of the Droplet Size Evolution in Flowing Immiscible Polymer Blends

**DOI:** 10.3390/polym11050761

**Published:** 2019-04-30

**Authors:** Ivan Fortelný, Josef Jůza

**Affiliations:** Institute of Macromolecular Chemistry of the Czech Academy of Sciences, Heyrovského Náměstí 2, CZ 162 06 Praha 6, Czech Republic; fortelny@imc.cas.cz

**Keywords:** polymer blends, phase structure, droplet breakup, coalescence, interfacial tension

## Abstract

Control of the phase structure evolution in flowing immiscible polymer blends during their mixing and processing is fundamental for tailoring of their performance. This review summarizes present state of understanding and predictability of the phase structure evolution in flowing immiscible polymer blends with dispersed structure. Results of the studies of the droplet breakup in flow, important for determination of the droplet breakup frequency and of the size distribution of the daughter droplets, are reviewed. Theories of the flow-induced coalescence providing equations for collision efficiency are discussed. Approximate analytic expressions reliably describing dependence of the collision efficiency on system parameters are presented. Available theories describing the competition between the droplet breakup and coalescence in flow are summarized and approximations used in their derivation are discussed. Problems with applicability of available theories on prediction of the droplet size evolution during mixing and processing of immiscible polymer blends, which have not been broadly discussed so far, are addressed.

## 1. Introduction

End-use properties of immiscible polymer blends are strongly affected by their phase structure, which is formed during their preparation and processing. Most polymer blends are prepared by melt mixing. Therefore, a reliable description of the phase structure evolution in immiscible polymer blends during their melt mixing is a necessary condition for tailoring their properties. However, description of the phase structure evolution is not an easy task; it can hardly lead to correct quantitative prediction of the blend morphology at the end of mixing. Flow field in mixing devices is complex and position-dependent even for the steady flow of homogeneous materials. At the present state of the art, it is impossible to combine a realistic description of this flow field with a description of microrheological events controlling the phase structure evolution. Therefore, real flow fields in mixing devices have to be replaced by simplified models.

An important group of polymer blends are blends with droplets in matrix morphology (e.g., plastics with impact strength enhanced by the addition of elastomers). For this type of blends, micro-rheological events controlling the size of dispersed droplets are qualitatively understood. It is commonly accepted that the size of dispersed droplets in flow is controlled by the competition between their breakup and coalescence [1,2,3,4]. Steady phase structure, i.e., time-independent droplet size distribution, is achieved after long enough steady flow of the blend when the blend components degradation is avoided. It relates to long enough mixing of a blend with properly stabilized components in batch mixers. Steady droplet size is usually not achieved during mixing in extruders and time evolution of the droplet size must be considered in this case.

For a discrete model of a system (droplet volume is a product of an integer and elementary volume *V*_1_), the following Equation describes change in the number of droplets of volume *kV*_1_, *n_k_* with time *t* [2]:(1)$$\frac{dn_{k}}{dt}=\frac{1}{2}\sum_{i+j=k}C(i,j)n_{i}n_{j}-F(k)n_{k}-\sum_{j=1}C(k,j)n_{k}n_{j}+\sum_{j=k+1}\omega (k,j)n_{f}(j)F(j)n_{j},$$
where *C*(*i*, *j*) is coagulation kernel, *F*(*i*) is overall breakup frequency, *n*_f_(*i*) is number of fragments formed at breakup of a droplet of volume *iV*_1_, and ω(*i*, *j*) is the probability that a fragment formed by the breakup of a droplet of volume *jV*_1_ will have volume *iV*_1_. Generally, *C*(*i*, *j*) should reflect the effect of further droplets on coalescence and terms describing simultaneous collisions of larger number of droplets should be added to Equation (1). Integro-differential equation, analogical to Equation (1), substitutes the set of equations for individual *n_k_* if continuous changes in droplet volumes are assumed. The left-hand side of Equation (1) is equal to zero for the steady state because the droplet size distribution is time-independent.

Functions *C*, *F*, *n*_f_, and ω are needed for solution of the set of Equation (1) or the relating integro-differential equation. Description of the breakup and coalescence of viscoelastic droplets in a viscoelastic matrix in complex flow fields is an extremely difficult task. Therefore, various approximate approaches to the description of the droplet breakup, coalescence and competition between them have been developed. Theoretical and experimental results of the studies of these events have been summarized in a number of reviews [1,2,3,4,5,6,7,8,9].

The aim of this paper is to present theories of the droplet breakup, coalescence, and the competition between them which can be utilized for understanding and evaluation of the phase structure formation in polymer blends. The further aim is a discussion of the plausibility of available approximate theories for the prediction of the droplet size formed by mixing and processing of polymer blends.

## 2. Droplet Breakup

### 2.1. Critical Capillary Number 

Deformation and breakup of droplets in flow have been an object of many theoretical and experimental studies, which cannot be all cited here. Most conclusive results have been obtained for Newtonian droplets in a Newtonian matrix in simple shear and extensional flows. Taylor [10] showed that the droplet deformation was controlled by the competition between the flow stress and the interfacial stress, equal to the Laplace pressure. For a spherical droplet, Laplace pressure is equal to 2*σ*/*R*, where *σ* is interfacial tension and *R* is droplet radius. Therefore, the droplet deformation can be expressed as a function of dimensionless capillary number *Ca* which is for shear flow defined as:(2)$$Ca=\frac{\eta_{m}\dot{\gamma }R}{\sigma },$$
where *η*_m_ is the viscosity of the matrix and $\dot{\gamma }$ is the shear rate. For other flow fields, *Ca* can be defined analogically to Equation (2), using the relevant component of the rate of deformation tensor. The most important parameters characterizing the droplet breakup are the critical capillary number, *Ca*_c_, i.e., the lowest *Ca* at which the breakup can occur, the breakup time *t*_B_, i.e., time necessary for droplet deformation from spherical shape to breakup, and the number and size of the formed droplet fragments. Generally, several types of the droplet breakup have been detected [3,4,5]. A decision, which of them is most relevant for mixing, processing, or measurements of rheological properties of a certain system, is not easy.

It has been found that *Ca*_c_ for Newtonian systems is a function of the ratio *p* of viscosities of the dispersed phase, *η*_d_, and matrix, *η*_m_, only [10,11,12,13]. Results of theoretical [10,11,12,13] and experimental studies [10,14,15,16,17,18] lead to the conclusion that the dependences of *Ca*_c_ on *p* for shear and extensional flows differ substantially. For shear flow, *Ca*_c_ has a minimum for 0.1 < *p* < 1, it steeply increases with increasing *p* for *p* > 1, and goes to infinity at about *p* = 4. For *p* < 0.1, *Ca*_c_ somewhat increases with decreasing *p*. In extensional flow, a minimum for *Ca*_c_ was observed at about *p* = 1. A very slight increase in *Ca*_c_ with increasing or decreasing *p* was observed for *p* > 1 or *p* < 1, respectively. The minimum value of *Ca*_c_ is substantially larger for shear than for extensional flow. The empirical equation, based on the results of experimental studies [10,14,15,16,17,18], for *Ca*_c_ as a function of *p* was proposed for shear flow [19] is:(3)$$\log Ca_{c}=-0.506-0.0995\log p+0.124{\left(\log p\right)}^{2}-\frac{0.115}{\log p-log4.08}.$$

Two empirical equations can be found for extensional flows. Utracki and Shi proposed [6]:(4a)$$\log Ca_{c}=-0.64853-0.02442\log p+0.02221{\left(\log p\right)}^{2}-\frac{0.00056}{\log p-0.00645}.$$

Peters et al. [20] used:(4b)$$\log Ca_{c}=0.0331{\left(\log p-0.5\right)}^{2}-0.699.$$

Equations (3) and (4) are frequently used in discussions of experimental data. It should be mentioned that Equation (3) is plausible for *p* < 4 only. Equation (4a) has a singularity for log *p* = 0.00645 and, therefore, it is not applicable for *p* slightly above 1. Equation (4b), although less frequently used than Equation (4a), does not contain a singularity and seems to be more convenient. The dependence of *Ca*_c_ on *p* described by Equation (3) relates to the breakup to two principal daughter droplets, possibly with odd numbers of small satellite droplets. For *p* << 1, the “tip streaming” mechanism is operative, where very small droplets break off the pointed ends of deformed original droplets [14,21]. *Ca*_c_ for tip streaming is independent of *p* and is equal to the minimum *Ca*_c_ for the breakup of the main droplet.

For polymer blends, *Ca*_c_ is a function of parameters of elasticity of the droplets and matrix besides *p*. Moreover, *η*_d_ and *η*_m_ are functions of the shear rate. Van Oene proposed to consider the effect of elasticity of the blend components on the phase structure evolution by substitution of equilibrium interfacial tension *σ* with its effective value *σ*_ef_, defined as [22]:(5)$$\sigma_{ef}=\frac{1}{6}R(N_{1,d}-N_{1,m}),\;$$
where *R* is the droplet radius and *N*_1,d_ and *N*_1,m_ are the first-normal stress differences of the droplets and matrix, respectively. A further concept of description of the effect of elasticity of the system components on droplet breakup is based on studies of Sundararaj et al. [4,23,24]. The concept is based on the assumption that the deforming forces can be expressed as *η*_m_$\dot{\gamma }$ + 2*G*’_m_, where *G*’_m_ is the storage modulus of the matrix. The deformation-resisting force is equal to *σ*/*R* + 2*G*’_d_, where *G*’_d_ is the storage modulus of the dispersed phase. The critical droplet radius for breakup, *R*_c_, is given by the dynamic equilibrium between the deforming and deformation-resisting forces. Therefore, *R*_c_ is given by the equation:(6)$$R_{c}=\frac{\sigma }{\eta_{m}\dot{\gamma }-2(G{’}_{d}-G{’}_{m})}.$$

The use of effective interfacial tension or the above concept of the dynamic equilibrium between the deforming and deformation-resisting forces for viscoelastic systems leads to the same qualitative conclusion: *Ca*_c_ increases with *N*_1,d_ (*G*’_d_) but decreases with increasing *N*_1,m_ (*G*’_m_) of the matrix with respect to the related Newtonian system.

Results of experimental studies of the effect of elasticity of the blend components on droplet breakup are summarized in [4]. Model studies have frequently been focused on blends containing Boger fluids where viscosity of the components is independent of the shear rate. Results of substantial part of experimental studies [23,24,25,26,27,28,29,30,31,32,33,34] are in qualitative agreement with the conclusions following from the above theoretical considerations. They show that the matrix elasticity enhances droplet deformation, causing the droplet breakup at lower *Ca*. The elasticity of droplets hinders their deformation, causing their breakup at higher *Ca*. On the other hand, there are some studies with results which are not in agreement with the above statements. Milliken and Leal [35] found that viscoelastic droplets showed greater deformation than Newtonian ones with equivalent *p* in a Newtonian matrix fluid under two-dimensional elongational flow at given *Ca*. Utracki and Shi [6] reported that droplet elasticity reduced its deformation for *p* < 0.5 but enhanced it for *p* > 0.5. Sibillo et al. [36] found that the breakup of a Newtonian droplet was hindered by the matrix elasticity in shear flow for *p* = 2; 0.6 and 0.04. This result is supported also by results of Flummerfelt [37] and Guido et al. [38]. Verhulst et al. [39] found that matrix elasticity reduced steady deformation and promoted the orientation of a Newtonian droplet in the shear flow. Generally, elasticity of droplets and matrix affects the shape of deformed droplets and type of the breakup mechanism [4,23,32,40]. A discrepancy among experimental results obtained so far indicates that interrelations between elastic properties and *p* can affect mechanism of the droplet breakup and, possibly, also *Ca*_c_.

It should be mentioned that the determination of *p* is not unambiguous for blends with other than Newtonian and/or Boger fluid components. Viscosity of the blend components, which reflects their structure in the flowing blend, should be considered. Therefore, their viscosity at a constant shear stress, equal to the average shear stress in a blend, seems to be most convenient for the calculation of *p*. The reason is that the stress is continuous at the interface of blend components in contrast to the rate of deformation. So far, the effect of other droplets on a droplet breakup has been studied quite rarely. Choi and Schowalter [41] extended Cox’s theory [12] to describe the droplet deformation in moderately concentrated emulsions of Newtonian liquids. They found a larger deformation of droplets due to an increase in viscosity of the emulsion above the viscosity of the matrix. Jansen et al. [42] proposed to replace viscosity of the matrix by viscosity of the emulsion in the definition of *p* and *Ca*. This mean-field concept satisfactorily explained experimental results but slightly overpredicted the stress at which breakup occurred. The mean field concept is used also in other studies of the phase structure evolution [43,44]. However, it should be mentioned that calculation of the blend viscosity using equation for emulsions is not generally approved due to a strong effect of the slip at the interface on the viscosity in many polymer blends [45].

### 2.2. Breakup Mechanisms

Besides *Ca*_c_, the breakup mechanism and its quantitative characteristics, i.e., the breakup time and number and size distribution of daughter droplets formed by the breakup, are necessary for solution of Equation (1). Two droplet mechanisms: stepwise (repeated breakup of a droplet into two halves) and transient (deformation of a droplet into a long slender filament followed by its breakup into a line of small daughter droplets) are considered as the basic ones. It is assumed that the stepwise breakup mechanism is operative for *Ca* not much larger than *Ca*_c_ and the transient one for *Ca* >> *Ca*_c_ [6]. Other breakup mechanisms, shown in Figure 1 and briefly discussed below, were found experimentally. 

The description of the stepwise mechanism of breakup of Newtonian droplets in a Newtonian matrix is based on the theories of Taylor [10] and his followers [11,12,13]. Taylor [10] proposed that breakup appears when the droplet deformation, *D*, achieves the value 0.5. *D* is defined as:(7)$$D=\frac{L-B}{L+B},$$
where *L* and *B* are the length and width of deformed droplets, respectively. *D* can be calculated from the theories describing droplet deformation [10,11,12,13]. For a system without coalescence where breakup into two halves without small satellite droplets is operative, steady droplet radii should lie in the interval (*R*_c_/2^1/3^, *R*_c_) where *R*_c_ for the shear flow is defined by:(8)$$R_{c}=\frac{\sigma Ca_{c}}{\eta_{m}\dot{\gamma }}.$$

It follows from Equation (8) that the dependence of steady droplet size on system parameters is controlled by the related dependence of *Ca*_c_.

A further parameter that should be known is the breakup time of a droplet, *t*_B_. It is the time needed for the deformation of a droplet from its initial spherical shape above the critical *D* and following droplet breakup. Experimental determination of *t*_B_ is somewhat more difficult than of *Ca*_c_. Cox [12,46] derived the following expression for time dependent *D* in steeply starting plane hyperbolic flow with velocity **v** = ($\dot{\gamma }$_h_*x*, −$\dot{\gamma }$_h_*y*, 0):(9)$$D(t)=2Ca_{h}\frac{19p+16}{16p+16}\left[1-\exp\left\{-\frac{20}{19pCa_{h}}{\dot{\gamma }}_{h}t\right\}\right]$$
with:(10)$$Ca_{h}=\frac{\eta_{m}R{\dot{\gamma }}_{h}}{\sigma }.$$

Using Taylor’s assumption that the droplet breakup appears at *D* = 0.5, the equation below follows from Equation (9) for *t*_B_:(11)$$t_{B}=-\frac{19pCa_{h}}{20{\dot{\gamma }}_{h}}\ln\left(1-\frac{1}{4Ca_{h}}\frac{16p+16}{19p+16}\right).$$

For uniaxial extensional flow with **v** = (−($\dot{\varepsilon }$/2)*x*, −($\dot{\varepsilon }$/2)*y*, $\dot{\varepsilon }$*z*) [46]:(12)$$D(t)=\frac{3Ca_{e}}{2}\frac{19p+16}{16p+16}\left[1-\exp\left\{-\frac{20}{19pCa_{e}}\dot{\varepsilon }t\right\}\right]$$
with:(13)$$Ca_{e}=\frac{\eta_{m}R\dot{\varepsilon }}{\sigma },$$
(14)$$t_{B}=-\frac{19pCa_{e}}{20\dot{\varepsilon }}\ln\left(1-\frac{1}{3Ca_{e}}\frac{16p+16}{19p+16}\right).$$

It follows from Equations (11) and (14) that *t*_B_ for planar hyperbolic and uniaxial extensional flows decreases with increasing related capillary number and deformation rate. These equations also define *Ca*_c_ by the condition that argument of logarithm is 0 for *Ca*_h_ or *Ca*_e_ is equal to *Ca*_c_.

In contrast to the plane hyperbolic and uniaxial extensional flows, *D*(*t*) is a periodic function of time with damping amplitude for shear flow according to Cox first order perturbation theory [12]. Theories considering the first order corrections to the spherical shape of droplets only fail to predict the droplet breakup in shear flow for systems with 0.1 < *p* < 3.6, determined experimentally. The second-order theory of Barthès–Biesel and Acrivos [13] yields to a breakup criterion but does not provide *t*_B_. 

Experimental data summarized by Grace [14] for shear flow show that reduced breakup time, defined as *t*_B_*σ*/(*R*_c_*η*_m_), is proportional to *p* (see Figure 2) and decreases with *Ca*/*Ca*_c_. It was also found that the droplet draw ratio at breakup, *L*_B_/2*R*, increased with *Ca*/*Ca*_c_ and grew with decreasing *p* for *p* < 1. The number of droplet fragments after its breakup increases with *Ca*/*Ca*_c_. Experimentally-determined dependence of the reduced breakup time for irrotational flows (extensional and plane hyperbolic) on *p* is similar to that for shear flow [14].

Cristini et al. [47] studied the droplet breakup in an impulsively started shear flow using a combination of numerical simulations and experiment. They found that the reduce breakup time scaled as:(15a)$$t_{B}\dot{\gamma }\propto {\left(\frac{R}{R_{c}}-1\right)}^{-1/2}$$
for *R*/*R*_c_ > 1 (see Figure 3). Fortelný and Jůza [48] proposed the following equation for $\dot{\gamma }$*t*_B_ using discussion in the Section VI in [47]:(15b)$$\dot{\gamma }t_{B}=(1+p)\left[\frac{k_{0}}{{\left(R*-1\right)}^{1/2}}+k_{1}{\left(R*-1\right)}^{1/2}\right],$$
where *R** is reduced droplet radius, i.e., the ratio *R*/*R*_c_, and *k*_0_ and *k*_1_ are dimensionless parameters.

The results of simulation and experimental studies show that the size of daughter droplets increases with the radius of parents droplets, *R*_0_, till about *R*_0_/*R*_c_ = 1.4. After that, the radii of daughter droplets are independent of *R*_0_ and scaled with *R*_c_. Unfortunately, the graph of the dependence of the radius of daughter droplet on *R*/*R*_c_ − 1 (Figure 7 in [47]) shows a certain variance and construction of the dependence of the number of daughter droplets on *R*/*R*_c_ − 1 is not easy. It was also found that the size of satellite droplets scaled with *R*_c_. Three size classes of satellite droplets were found experimentally: (1) 0.4 < *R*/*R*_c_ < 0.8, (2) 0.1 < *R*/*R*_c_ < 0.3, and (3) *R*/*R*_c_ < 0.1.

For *R*/*R*_c_ >> 1, divergence of *t*_B_$\dot{\gamma }$ with *R*/*R*_c_ was found [47,49]. However, different powers of *R*/*R*_c_ were found by Van Puyvelde et al. [49] and Cristini et al. [47]. For *R*/*R*_c_ >> 1, the droplets in fact disintegrate by the Tomotika mechanism [50]. The description of the breakup is based on the assumption that a droplet is almost affinely deformed into a long, slender filament under this condition. When the radius of the filament decreases to the value for which the flow field stress and the interfacial stress attain the same order of magnitude, small disturbances at the interface of the filament grow and finally result in the breakup of the filament into a chain of small droplets (see the second row in Figure 1). The growth of disturbances on a Newtonian thread in a quiescent Newtonian continuous phase was described by Tomotika [50]. His theory was generalized by Palierne and Lequeux [51] to a viscoelastic thread in a quiescent viscoelastic continuous phase.

It was found that distortions with a wavelength *λ* greater than the initial circumference of the cylinder would grow exponentially with time [1,2,6,50]:(16)$$\alpha (t)=\alpha_{0}e^{qt},$$
where *α*(*t*) is distortion amplitude at time *t*, *α*_0_ is an initial distortion amplitude, and *q* is the growth rate. At the beginning, small amplitudes of all wavelengths are present. For a certain value of *p*, one disturbance with dominant wavelength, *λ*_m_, grows most rapidly and causes disintegration of the cylinder. For the breakup time, *t*_B_, a Newtonian cylinder with initial radius *R*_0_ in quiescent Newtonian matrix with viscosity *η*_m_, the following equation was derived [50,51]:(17)$$t_{B}=\frac{2\eta_{m}R_{0}}{\sigma \Psi }\ln\frac{0.82R_{0}}{\alpha_{0}},$$
where *σ* is interfacial tension and Ψ(*λ*, *p*) is a decreasing function of *p* and can be determined from equations in [50,51] or from graphs presented in [7]. *α*_0_ was estimated assuming that it is caused by temperature fluctuations due to Brownian motion [52]. However, experimental data better match the order of magnitude of larger *α*_0_, and *α*_0_ is mostly considered as an adjustable parameter [7]. Equation (17) shows that the breakup time increases with *η*_m_, *p*, and *R*_0_, and decreases with *σ*. Experimental studies confirmed that Tomotika’s theory describes Newtonian systems satisfactorily, at least for small and moderate *α* [7]. The effect of the elastic properties of a thread and matrix is discussed in [8]. An increase in relaxation times of the matrix and thread was found to enhance the rate of growth instability. Experimental results for some systems with viscoelastic components agree quite well with predictions of Tomotika’s theory [7]. For others, especially polymer solutions, differences from sinusoidal disturbances were found [7].

The breakup of an extending liquid cylinder immersed in another flowing immiscible liquid was analyzed in several papers [8,53,54]. The results are summarized in [7]. The cylinder is continuously stretched in flowing matrix. It was shown [54] that the relevant parameter of stretching is the orientation-dependent stretching rate $\dot{\varepsilon }$(=**E:mm**, i.e., the scalar product of the rate of strain tensor **E** and of the dyadic product of the orientation vectors **m**). Due to the stretching of the thread, the wavelength of a disturbance and the thread radius change as:(18)$$\lambda =\lambda_{0}e^{\dot{\varepsilon }t},$$
(19)$$R=R_{0}e^{-\dot{\varepsilon }t/2},$$
where *λ*_0_ is the value of *λ* at *t* = 0. Evolution of *α* can be found by solution of the balance equation of mass and motion of an elongated thread with Newtonian behavior and prescribed boundary conditions [7]. The following equation was derived for the amplitude of a disturbance:(20)$$ln\frac{\alpha }{\alpha_{0}}=\int_{x_{0}}^{x}\left[\frac{-x_{0}^{1/3}(1+x^{2})}{3x^{4/3}R_{0}\eta_{m}\dot{\varepsilon }/\sigma }\Phi (x,p)+\frac{p-1}{x}\bar{\Phi }(x,p)+\frac{1}{3x}\right]dx,$$
where *x* = 2π*R*/*λ* and *x*_0_ = 2π*R*_0_/*λ*_0_. The Φ(x,p) and $\bar{\Phi }(x,p)$ functions are defined in [7] or [53]. According to Equation (20), *α* is damped at first, then it grows for a while, and finally continues to be damped. During the first damping stage, the amplitude remains equal to *α*_0_ because it cannot damp below its initial “noise” level. The second stage starts at time *t*_s_, relating to a local minimum of Equation (20). The value of *α* at *t* > *t*_s_ can be obtained by integration of Equation (20) from *x* corresponding to *t*_s_ with the condition *α*(*t*_s_) = *α*_0_. In the third stage, the amplitude damps again, but in reality, it cannot decrease below *α*_0_. The breakup occurs when *α*(*t*) = *R*(*t*). A more detailed discussion of the droplet breakup by Tomotika’s mechanism in flow can be found in [7]. The breakup time of the elongated thread is a sum of *t*_s_ and of time, *t*_g_, needed for the growth of *α* to its critical value. It should be mentioned that the time needed for deformation of a drop into an elongated thread should be added to the breakup time of the thread if the whole time from the start of system deformation to the breakup is considered. For this case, the following relation was proposed by Van Puyvelde et al. [49]:(21)$$t_{B}\propto {\dot{\gamma }}^{-1}Ca^{2/3}.$$

A comparison of the results for an elongated thread immersed in a quiescent and a flowing immiscible liquid leads to the conclusion that the breakup time increases and the size of fragments decreases in stretching flow with respect to the quiescent state [7]. For flow with constant $\dot{\varepsilon }$ (various types of elongational flows), the ratio of the radius of formed droplets, *R*_f_, to *α*_0_ decreases with increasing *p* and *η*_m_$\dot{\varepsilon }$*α*_0_/*σ*. For values of *η*_m_$\dot{\varepsilon }$*α*_0_/*σ* not too large, the relation *R*_f_ ∝ (*η*_m_*η*_d_)^−0.45^ is valid. It should be mentioned that *R*_f_ can be smaller than *R*_c_/2^1/3^, i.e., the smallest size of droplets formed by the breakup of a droplet into two halves. For steady, simple shear flow, the results are more complicated [7,54]. Generally, the decrease in *R*_f_ with increasing $\dot{\gamma }$ is less steep, and its dependence on *p* is less pronounced than in extensional flows. Theoretically predicted trends matched experimental results quite well. However, larger sizes of formed droplets were found than those predicted by theory.

The number of daughter droplets can be determined from the knowledge of their radius, *R*_d_. The following equation was proposed for by Van Puyvelde et al. [49] for radius of the thread, *r*_0_, at *Ca* >> *Ca*_c_:(22)$$r_{0}=R{\left(\dot{\gamma }t_{0}\right)}^{-1/2}.$$

Radius *R*_d_ of droplets formed by breakup of the fiber with radius *r*_0_ is determined for incompressible liquids by the equation [55]:(23)$$R_{d}=r_{0}\sqrt[{3}]{\frac{3\pi }{2x_{m}}}=r_{0}f(p)$$
where *x*_m_ is the dominant wave number, which is a function of *p*. Equations (22) and (23) lead to the following equation for the number of daughter droplets, *n*_f_, if volume of possible satellite droplets is neglected:(24)$$n_{f}=\frac{1}{2f^{3}(p)}\frac{R\eta_{m}\dot{\gamma }}{\sigma }=\frac{Ca}{2f^{3}(p)}$$

A further possible breakup mechanism is end pinching—two droplets (larger than that formed by the Tomotika mechanism)—are formed at the ends of a finite elongated drop and tear off [5]. After that, the “neck” either continues to breakup in the same way or relaxes in shape. The mechanism is typical of the step changes in flow, e.g., flow cessation. The mechanism was detected experimentally [10,14,56] and explained as a consequence of an interfacial tension-driven flow associated with curvature variations along the surface of finite drop [5]. Experimental study and mathematical modeling show that the breakup by this mechanism is suppressed for *p* >> 1, where relaxation of a highly deformed drop to a spherical shape is possible, because high droplet viscosity damps internal flows. High capillary numbers are necessary to cause a transient elongation because very elongated steady shapes are possible for these systems. Formation of small satellite droplets and of still smaller subsatellite droplets was found experimentally [5,9,14] (see the third row in Figure 1). End pinching appears for moderate droplet deformation—not large enough for development of Tomotika’s mechanism. For small droplet deformation, end pinching passes to stepwise mechanism, possibly with formation of satellite droplets.

Tip streaming of small droplets from the sharp ends of deformed mother drop appears in systems with *p* << 1 (see the fourth row in Figure 1). In this case, a number of daughter droplets, more than order of magnitude smaller than mother drop, are formed. Other mechanisms were detected in viscoelastic systems [4]. During parallel breakup, the droplet stretches into a thin sheet along the flow direction and bursts into two or more daughter droplets and one or several satellite droplets (see the fifth row in Figure 1). During surface erosion (the sixth row in Figure 1), thin ribbons tear off the drop surface; the ribbons further break up into a chain of small droplets. Surface erosion was observed in systems with *p* >> 1. During vorticity alignment and breakup, the droplets are elongated in the vorticity direction and subsequently break up (the seventh row in Figure 1).

It follows from the above that information about the breakup time and a number and size distribution of the droplet fragments is still limited. Moreover, the breakup time and number of the fragments are different for various breakup mechanisms. Therefore, recognition of a decisive breakup mechanism is essential for construction of reliable expression for breakup frequency in Equation (1). A necessary condition is an estimation of the ratio of average droplet radius to *R*_c_ for a given system. 

## 3. Coalescence

Flow-induced coalescence is a consequence of droplet collisions brought on by the difference in velocity. Its scheme is shown in Figure 4. While droplets approach each other, their hydrodynamic interaction and molecular forces between them increase. Coalescence can be split into four steps [57]:approach of the droplets;drainage of the continuous phase trapped between the droplets, possibly deformed by the axial force;rupture of remainder of the continuous phase, usually by the formation of a “hole” on the thinnest spot; andevolution of a “neck” between droplets and formation of a coalesced droplet.

Usually, it is assumed [46,57] that the coalescence is controlled by the first two steps. The rupture of the remainder of the continuous phase is assumed to be much faster than the other steps [57]. Possibility of the neck breakup during its evolution is mostly not considered.

Generally, the rate of coalescence, *J*, can be calculated from the flux of a pair of droplets through the collision surface; this flux is equal to the flux at *r* → ∞ through a cross-section *A*_c_ (upstream interception area) [58,59,60]:(25)$$J=-n_{i}n_{j}\int_{A_{c}}v_{0}\cdot ndS,$$
where *n*_i_ and *n*_j_ are the number of droplets with radii *R*_i_ and *R*_j_, respectively, **v_0_** is the flow velocity unperturbed by the presence of the droplets, **n** is the outward unit normal to the spherical contact surface, and d*S* is the surface element. If any interaction between droplet before touching is neglected (Smoluchowski approximation), the following equations can be derived from Equation (25) using geometrical considerations for shear flow [61]:(26)$$J_{0}=\frac{4}{3}{\left(R_{i}+R_{j}\right)}^{3}\dot{\gamma }n_{i}n_{j}$$
and for uniaxial extensional flow [58]:(27)$$J_{0}=\frac{8\pi }{3\sqrt{3}}{\left(R_{i}+R_{j}\right)}^{3}\dot{\varepsilon }n_{i}n_{j}.$$

Coalescence is characterized by the collision efficiency, *P*_c_, defined as the ratio of real rate of coalescence to the rate of coalescence calculated using Smoluchowski approximation, i.e.,
*P*_c_ = *J*/*J*_0_(28)

Generally, two types of forces between colliding droplets exist: hydrodynamic interaction and attractive molecular forces (van der Waals in typical polymer blends). *P*_c_ can be larger than 1 if strong enough attractive forces between droplets exist. However, attractive molecular forces between droplets are weak and short-range in most polymer blends. The hydrodynamic interaction, leading to a decrease in *P*_c_, prevails for them. Therefore, *P*_c_ can be interpreted as the probability that the collision of droplets, in a geometrical sense (Smoluchowski approximation), is followed by their fusion.

Description of the drainage of matrix trapped between colliding droplets is key problem in the calculation of *P*_c_. This task is complicated. Radial and axial components of the driving force of coalescence cause the droplet pair to rotate around their center of inertia and approach each other, respectively. These components change rapidly during the coalescence even if the magnitude of the driving force is almost constant. Magnitude of the axial component of the driving force affects the shape and extent of deformed part of colliding droplets. On the other hand, these parameters affect the rate of droplet approach and rotation. Therefore, the self-consistent problem should be solved when describing the matrix drainage. As a consequence, only few of a very large number of papers describing coalescence deal with calculation of *P*_c_, i.e., consider the competition between the approach and rotation of colliding droplets [62]. 

The following equation was derived for radius of the flattened part of a droplet, *r*_f_, [57]:(29)$$r_{f}={\left(\frac{RF_{c}}{2\pi \sigma }\right)}^{1/2},$$
where *F*_c_ is the driving force of the coalescence given by the difference between velocities of colliding droplets and their friction resistances. More recent studies have shown that dimple shapes, not flattened areas, are typical of colliding droplets [62]. However, we believe that Equation (29) can still serve for estimation of the extent of droplet deformation, which is essential for the choice of the equations describing the matrix drainage between colliding droplets. It should be mentioned that *F*_c_ and, therefore, *r*_f_ change during a course of the coalescence due to droplet rotation. Only in the case when the maximum *r*_f_ is small, i.e., smaller or comparable with the critical distance, *h*_c_, for the breakup of remainder continuous phase trapped between the colliding droplets, coalescence of the spherical droplets can be considered.

The equation describing approach of Newtonian spherical droplets in a Newtonian matrix for the short inter-droplet distances was derived by Zhang and Davis [63]:(30)$$-\frac{dh}{dt}=\frac{2F_{c}h}{3\pi \eta_{m}R^{2}g(m)},$$
where *h* is the distance between droplet surfaces and:(31)$$g(m)=\frac{1+0.402m}{1+1.711m+0.46m^{2}}$$
and:(32)$$m=\frac{\eta_{m}}{\eta_{d}}{\left(\frac{R}{2h}\right)}^{1/2}.$$

In addition to the equation for matrix drainage between spherical droplets, equations for approach of plane surface assuming that *r_f_* >> *h* were also derived for fully mobile (*p* << 1), partially mobile (*p* ≈ 1), and immobile (*p* >> 1) interfaces [2,57,64]. Unfortunately, these equations do not converge one to another with the change in *p* and boundaries of applicability of individual equations are not established. Jeelani and Hartland [65] derived equation for the plane surface approach plausible for finite *p*, which passes to the equation for immobile interface for *p* → ∞:(33)$$-\frac{dh}{dt}=\frac{8\pi \sigma^{2}h^{3}}{3\eta_{m}R^{2}F_{c}}\left(1+3C\frac{\eta_{m}}{\eta_{d}}\right),$$
where *C* is the ratio of the circulation length and droplet distance and is of the order of 1. Fortelný and Jůza [66] assumed that *C* was an arbitrary function of the system parameters. *C* meets the following conditions: Equation (33) passes to the equation for fully mobile interface for *p* → 0 and to the equation for immobile interface for *p* → ∞. Following these assumptions, Equation (34) was derived for arbitrarily mobile interface:(34)$$-\frac{dh}{dt}=\frac{8\pi \sigma^{2}h^{3}}{3\eta_{m}R^{2}F_{c}}\left\{1+\frac{3a}{p}\left[1-exp\left\{-\frac{p}{3a}\left(\frac{RF_{c}}{4\pi h^{2}\sigma }-1+bp\right)\right\}\right]\right\},$$
where *a* and *b* are adjustable dimensionless parameters. It was found that using Equation (34) with parameters *a* = 8 and *b* = 1 for calculation of *P*_c_ led to satisfactory agreement with both the theory of Rother and Davis [60] and experimental data [66]. It should be mentioned that, for colliding droplets with different radii *R*_1_ and *R*_2_, *R* in Equations (30)–(34) should be substituted by equivalent radius *R*_eq_ defined as:(35)$$\frac{1}{R_{eq}}=\frac{1}{2}\left(\frac{1}{R_{1}}+\frac{1}{R_{2}}\right)\Rightarrow R_{eq}=2\frac{R_{1}R_{2}}{R_{1}+R_{2}}.$$

Generally, driving force of the coalescence depends on *h*. As *P*_c_ is mostly controlled by the droplet motion at short inter-droplet distance and dependence of *F*_c_ on *h* is weak, contact force, calculated for touching spheres in flow is frequently considered [59,60,67]. For the shear-flow-induced coalescence, the driving force of the coalescence, *F*_s_, is described by the equation [60,67]:(36)$$F_{s}=\frac{1}{2}K(p,\Lambda )\pi \eta_{m}\dot{\gamma }R_{eq}^{2}\sin^{2}\theta \sin2\varphi ,$$
where Λ is the ratio of radii of smaller to larger droplets, *θ* is polar angle, *φ* is azimuth, and *K* can be expressed as:(37)$$K(p,\Lambda )=3\frac{{\left(1+\Lambda \right)}^{3}}{4\Lambda }D*(p,\Lambda ),$$
where the method of calculation of a dimensionless function *D** and its values for selected set of *p* and Λ can be found in [67]. For Λ = 1, the following equation for the dependence of *K* on *p* was proposed [68]:(38)$$K(p)=K_{\infty }\frac{2+3p}{3(1+p)},$$
where *K*_∞_ is 12.24 for touching spheres [69]. However, it should be mentioned, that values of *K* calculated from Equations (37) and (38) somewhat differ. For uniaxial extension, the driving force of the coalescence, *F*_e_, is described by:(39)$$F_{e}=K(p,\Lambda )\pi \eta_{m}\dot{\varepsilon }R_{eq}^{2}(1-3\cos^{2}\theta ).$$

Elmendorp and coworkers [46,70] recognized that *P*_c_ was controlled by the competition between the approach and rotation of colliding droplets. The droplets fuse if they approach to the critical distance *h*_c_ sooner than they rotate to the angle where *F*_c_ changes sign and starts moving away of the droplets. They assumed that the resistance against the approach of the droplets was the sum of resistance against drainage between flattened parts of droplets and that between approaching hard spheres, i.e.,
(40)$${\left(\frac{dh}{dt}\right)}^{-1}={\left(\frac{dh}{dt}\right)}_{f}^{-1}+{\left(\frac{dh}{dt}\right)}_{Sp}^{-1}.$$

Expression for the rate of approach of flattened droplets with mobile or immobile interface was substituted for the first term on the right-hand side of Equation (40). Equation (30) with *g*(*m*) = 1 (limit for *p* → ∞) was used for the second term on the right-hand side of Equation (40). By combining Equation (40) with equations describing the time dependence of rotational angles, the equation describing the dependence of *h* on rotational angles *φ* and *θ* was derived. Its solution determines the limit value of initial angles *φ*_0_ and *θ*_0_ (at the inter-droplet distance *h*_0_ at assumed start of the coalescence) for which *h* at *φ* = π/2 is smaller than *h*_c_. Equation (40) apparently overestimates the real resistance against the approach of colliding droplets [69]. This leads to the conclusion that the theory agrees with an experiment when the expression for mobile interface and *h*_c_ = 50 nm (unrealistic large value) are used.

The approach by Janssen and coworkers [7,43,64] is based on pre-averaging of both the driving force of the coalescence and the time of droplet rotation. The approach of flattened droplets is considered in this theory. It is assumed that the driving force of the coalescence of the droplet with the same radius, *R*, in the shear flow is given by the equation:(41)$$F_{c}=6\pi \eta_{m}\dot{\gamma }R^{2}.$$

The time of coalescence, *t*_c_, is determined by integration of equations for approach of flattened droplets with mobile, partially mobile, or immobile interfaces from *h*_0_ to *h*_c_ and from 0 to *t_c_* using Equation (41) for *F*_c_. *P*_c_ is calculated using the equation:(42)$$P_{c}=\exp\left\{-t_{c}/t_{i}\right\},$$
where interaction time *t*_i_ is defined as:(43)$$t_{i}={\dot{\gamma }}^{-1}.$$

This theory has been quite frequently used to evaluate experimental results because it provides explicit expressions for *P*_c_. Unfortunately, the equation for the approaching of flattened droplets leads to unphysical results for small *F*_c_, where colliding droplets keep almost spherical shapes.

Wang et al. [59] derived the theory of coalescence in shear, uniaxial extensional and compressional flows based on trajectory analysis. They assumed that the droplets maintain their spherical shape through the whole approach. Changes in hydrodynamic interaction during the droplet approach and rotation are considered. Van der Waals forces between droplets are included directly in equations describing droplet motion. Attractive molecular forces are included into the equation for the droplet approach and *h*_c_ = 0 is considered. Wang et al. calculated numerically the droplet trajectories and used them for the determination of *A*_c_ in Equation (25). They showed that, neglecting inter-droplet van der Waals forces, *P*_c_ is a decreasing function of *p* and an increasing function of the ratio, Λ, of the radius of a smaller droplet to the radius of a lager one. On the other hand, *P*_c_ is independent of *η*_m_, *R*_eq_ and the deformation rate ($\dot{\gamma }$ for shear and $\dot{\varepsilon }$ for extensional flows).

Rother and Davis [60] generalized the theory of Wang et al. [59] for deformable droplets, considering droplet deformation as a small but singular perturbation. Governing equations for flow in the thin matrix film and in the droplet in the vicinity of near contact are solved, taking into account radius of the droplet deformed part (cf. Equation (29)). For small capillary numbers, *Ca*, it was found that *P*_c_ was identical to that predicted in [59] for spherical droplets in the system with the same parameters. At a certain *Ca*, *P*_c_ starts decreasing steeply to a very low value. The dependence of *P*_c_ on *R* is similar to that of *P*_c_ on *Ca* [60], i.e., a steep decrease in *P*_c_ appears at a certain *R*_L_. It should be mentioned that *P*_c_ decreases with decreasing Λ for a polydisperse system. On the other hand, the value of the average droplet radius 〈*R*〉 (〈*R*〉 = (*R*_1_ + *R*_2_)/2) equal to *R*_L_ increases with decreasing Λ.

Fortelný and Jůza [71] derived a theory based on the assumption that the droplet approach could be described by the equation for spherical droplets (Equation (30)) at small *F*_c_ and the equation for deformed droplets (Equation (34) was chosen as the most reliable [66]) for large *F*_c_. That of these equations for d*h*/d*t* predicting slower droplet approach for a given *F*_c_ was always used. The equations derived by Wang et al. [59] for the rate of the droplet rotation in shear flow with **v_0_** = (−$\dot{\gamma }$*y*, 0, 0)
(44)$$\frac{d\phi }{dt}=\dot{\gamma }\left[\left(1-\beta /2\right)\sin^{2}\varphi +(\beta /2)\cos^{2}\varphi \right],$$
(45)$$\frac{d\theta }{dt}=-\dot{\gamma }(1-\beta )\sin\theta \cos\theta \sin\varphi \cos\varphi ,$$
and in extensional flow with **v_0_** = $\dot{\varepsilon }$(–*x*, –*y*, 2*z*):(46)$$\frac{d\theta }{dt}=-3(1-\beta )\dot{\varepsilon }\sin\theta \cos\theta$$
were used. The choice of **v_0_** for shear flow corresponds to positive driving force of coalescence for azimuth *φ* between 0 and π/2 [72]. *β* in Equations (44)–(46) is a function of *p* and Λ defined and calculated in [59,60,67]. Generally, *β* depends on *h*, but its limit value for *h* → 0 is considered for simplicity in calculations in [66,71,72,73]. Limit initial values of rotational angles, *φ*_0_ and *θ*_0_, for which the droplet collision is followed by their fusion, are calculated by combination of the equations for the rate of rotation (Equations (44) and (45) or Equation (46)) with relevant equations for the droplet approach. The condition is that *h* < *h*_c_ before achieving values of *φ* and *θ*, for which *F*_c_ changes sign. The determined *φ*_0_ and *θ*_0_ are used for calculation of *P*_c_ from Equations (25) and (28) [66,71,72,73].

Dependences of *P*_c_ on system parameters, calculated according to the theory described in the preceding paragraph, for coalescence of droplets with the same radii in systems with *h*_c_ << *R* are very similar to those calculated according to Rother’s and Davis’ theory [60]. For small *Ca*, *P*_c_ is a decreasing function of *p*; it is independent of other system parameters. At a certain *Ca*, *P*_c_ starts decreasing steeply to a negligible value. It should be mentioned that intersection of *P*_c_ calculated using Janssen’s theory [7,43,64] with *P*_c_ for spherical droplets, calculated with combination of Equation (30) with Equations (44) and (45), is very near to the value of *R* at which *P*_c_ calculated according to the theory from the preceding paragraph starts falling [71]. Analysis of the result of the Rother and Davis, and Fortelný and Jůza theories for coalescence of droplets with different radii showed that the dependences of *P*_c_ on 〈*R*〉 for monodisperse and polydisperse systems differed substantially [74]. *P*_c_ gradually decreases in a range of 〈*R*〉, the width of which increases with the polydispersity of the droplet radii. The dependence of *P*_c_ on the average droplet size for blends containing droplets with the same droplet radii and for those with droplets with broad radii distribution (denoted as polydisperse) is shown in Figure 5.

All the above described results have been obtained for Newtonian droplets in a Newtonian matrix. However, droplets and matrix in polymer blends are viscoelastic liquids. Attempts to develop theory of flow-induced coalescence in viscoelastic systems have been rare so far. Yu and Zhou [75] modeled shear flow-induced coalescence by the diffuse interface method. Their model of the interface is rigorous but the study does not consider complete geometry of the coalescence. Therefore, the theory does not provide *P*_c_ as a function of system parameters. Fortelný and Jůza [71] generalized their theory described above to systems with a viscoelastic matrix. They assume that the relation between drag force, **F**, and velocity of a particle, **u**, moving in the matrix can be utilized for coalescence, where the force inducing the droplet approach is caused by the difference between unperturbed velocities of colliding droplets and resistance to their approach is caused by their hydrodynamic interaction. The following equation was derived [76,77] for motion of the particle with friction resistance, *ζ*, in viscoelastic matrix, described by Maxwell model with relaxation time *τ*_m_:(47)$$F=\zeta u-\tau_{m}\frac{dF}{dt}.$$

In analogy with Equation (44) and Stokes’ law for Newtonian systems, *F*_c_ was substituted with *F*_c_ + *τ*_m_d*F*_c_/d*t* in Equations (30) and (34). In the first papers [66,71,73], modified equations for the droplet approach were combined with the equations for rotations of the droplet pairs in a Newtonian matrix.

Later [72], the equations for the droplet pair rotation were modified considering the effect of the matrix elasticity on the droplet rotation in the same way as on their approach. For a system with the Newtonian matrix, the relative velocity of colliding droplets with the same radius *R*, **v_12_**(**r**), can be expressed as [59,60]:(48)$$v_{12}(r)=\Omega \times r+E\cdot r-\left[A(s)\frac{rr}{r^{2}}+\beta (s)\left(I-\frac{rr}{r^{2}}\right)\right]\cdot E\cdot r,$$
where **Ω** is angular velocity tensor, **E** is rate of strain tensor, **r** is the vector from the center of droplet 1 to the center of droplet 2, **I** is the unit second-order tensor, and *s* = *r*/*R* is the dimensionless center-to-center distance and functions *A*(*s*) and *β*(*s*) are defined in [59,60]. It follows from the analysis of the relations between related radial components of **v_12_** and *F*_s_ and *F*_e_ that they are proportional through effective friction coefficient, *ζ*, expressed as:(49)$$\zeta =\frac{1}{2}K\pi \eta_{m}R.$$

Axial components of the driving force of the coalescence were calculated from Equation (48) using effective friction coefficient *ζ*. After that, Equation (47) was used for calculation of the relations between circumferential velocity of droplets in a viscoelastic matrix and axial components of the force causing their rotation. The limit of initial rotational angles *φ*_0_ and *θ*_0_ was determined and *P*_c_ was calculated by the above described procedure used for Newtonian systems. It has been found that, at least for relaxation times typical of commercial polymers and deformation rates for which substantial coalescence in Newtonian systems is assumed, the character of *P*_c_ vs. *R* dependence is similar to that of Newtonian systems. Matrix elasticity has a weak effect on the critical value of *R* where *P*_c_ starts decreasing steeply in the both shear and extensional flows. For extensional-flow-induced coalescence, a decrease in *P*_c_ with *τ*_m_ in the range of small *R* is predicted. The predicted decrease is stronger when the effect of the matrix elasticity on the droplet rotation is considered. For shear flow-induced coalescence, the theory neglecting the effect of *τ*_m_ on the droplet rotation predicts a slow decrease of *P*_c_ with *τ*_m_ in the range of small *R*. On the other hand, an increase in *P*_c_ with *τ*_m_ in this range of *R* is predicted by the theory considering the effect of the matrix elasticity on the droplet rotation. To the best of our knowledge, no theory dealing with the effect of the droplet elasticity on their coalescence has been proposed so far.

Flow-induced coalescence in polymer blends and model Newtonian emulsions has been frequently directly (breakup of the droplet has been excluded by the choice of experimental conditions) and indirectly (the droplet size distribution has been affected by the competition between the droplet breakup and coalescence) experimentally studied. All related papers cannot be cited here. Only some of the results plausible for verification of the conclusions, following from the above described theories, are cited below. We believe that, in spite of the approximations and uncertainties in modeling the droplet deformation during their collision [78,79,80,81,82,83] and neglecting the effect of diffuse interface [84], theories of Rother and Davis [60,85] and those of Fortelný and Jůza [66,71,72,73,74] reliably reflect main features of coalescence. Therefore, they should provide a plausible, at least semi-quantitative, description of the coalescence. *P*_c_ independent of *R*_eq_ and of the deformation rate is predicted by the both groups of theories for small *R*_eq_ where *r*_f_, given by Equation (29), is negligibly small. In this range of *R*, a decrease of *P*_c_ with increasing *p* and decreasing Λ is predicted. A decrease in *P*_c_ with *p* follows from experimental results in [86,87]. The statement that *P*_c_ decreases with decreasing Λ is supported by detected strong growth of polydispersity in the droplet size after the start of coalescence [88,89,90]. Existence of critical *R*_eq_, for which *P*_c_ at a given set of the system parameters falls to a negligible value, also predicted by the both groups of theories, is confirmed by a great deal of experimental results [86,87,88,89,90,91,92,93]. Experimentally determined dependence of critical *R*_eq_ on *p* [91] matches the arbitrarily mobile interface (AMI) model (Equation (34) for approach of deformed droplets).

As mentioned above, Janssen’s theory of coalescence [7,43,64], most frequently used for the evaluation of experimental data (e.g., [91,92,94,95,96,97]), provides critical *R* similar to that following from the theory using a switch between equations for the approach of spherical and deformed droplets [71], when the same equations for approach of deformed droplets are used. Therefore, Janssen’s theory can be used for reliable estimation of critical *R*. From the above discussion of available theories of the matrix drainage between deformed droplets it follows that the AMI model should be used to describe the approach of deformed droplets. *t*_c_ can be calculated by integration of Equation (34), where *F*_c_ is substituted from Equation (41). *P*_c_ is obtained by the substitution of calculated *t*_c_ and *t*_i_ from Equation (43) into Equation (42). For *h*_c_ << *h*_0_, *P*_c_ calculated by the above procedure is given by [66]:(50)$$P_{c}=\exp\left\{-\frac{9Ca^{2}R^{2}}{4\left(1+\frac{3a}{p}\right)}\left[\frac{1}{2h_{c}^{2}}+\frac{a}{pCaR^{2}}\ln\left(\frac{1+\frac{3a}{p}-\frac{3a}{p}\exp\left\{-\frac{p}{3a}\left(\frac{3CaR^{2}}{2ah_{c}^{2}}-1+bp\right)\right\}}{1+\frac{3a}{p}-\frac{3a}{p}\exp\left\{-\frac{p}{3a}\left(\frac{3CaR^{2}}{2ah_{0}^{2}}-1+bp\right)\right\}}\right)\right]\right\}.$$

It should be mentioned that Equation (50) is not applicable in the region of small *R*, where it underestimates value of *P*_c_ and neglects its dependence on *p*. Steep decrease of *P*_c_ in a narrow range of *R* following from Equation (50) is predicted also by the trajectory analysis considering the droplet deformation [60,85] and by the theory using switch between equations for approach of spherical and deformed droplets [66,71,73] for a pair of droplets (*R*_eq_ should be used instead of *R* for droplets with different radii). Similar dependence of *P*_c_ on average *R* can be assumed for systems with a low polydispersity [74]. In contrast, it follows from the above mentioned theories that *P*_c_ gradually decreases in a broad range of average *R* in systems highly polydisperse in the droplet size. In this case, the frequently used split of the plot of the average *R* vs. $\dot{\gamma }$ into regions, where breakup or coalescence or both of or none of them appears [70,89,94,95,96,97], does not capture the situation in a system.

Some approximations, not always consistent and well approved, were used in [71,72,73]. Their effect on *P*_c_ was analyzed [98]. It has been found that their removing affects magnitude of calculated *P*_c_ but keeps qualitative dependence of *P*_c_ on system parameters. However, using a more correct description of the droplet approach at medium distances leads to serious problems with the description of the coalescence in viscoelastic matrixes.

Theories of Rother and Davis [60] and those of Fortelný and Jůza [66,71,72,73,74] provide only numerical results for *P*_c_ even for monodisperse Newtonian droplets in a Newtonian matrix. Therefore, approximate expressions for *P*_c_(*R*) were proposed. Rother and Davis [85] suggested for *P*_c_(*R*) in shear flow:(51)$$P_{c}=\langle \begin{array}{cccc} P_{\mathrm{sph}} & & & \mathrm{for}\;R\leq R_{L,\mathrm{RD}}\\ P_{sph} & \frac{{\left(R_{L,RD}-R_{F}\right)}^{2}}{{\left(R-R_{F}\right)}^{2}} & & \mathrm{for}\;R>R_{L,\mathrm{RD}} \end{array}$$
where:(52)$$R_{L,RD}=0.420{\left[\frac{p+1.02\ln p+14.9}{p^{2}}{\left(\frac{p+1}{p+2/3}\right)}^{2}\frac{A\sigma^{3}}{{\left(\dot{\gamma }\eta_{m}\right)}^{4}}\right]}^{1/6}.$$

They approximated numerical results for *P*_sph_ [59] as [85]:(53)$$P_{sph}={\left[0.7949\;p+0.1061\ln p+1.8284\right]}^{-1}.$$

*R*_F_ was determined empirically by a numerical data fit for *P*_c_ vs. droplet radius. Another approximation, without necessity to fit numerical data for determination of the equation parameters, was proposed by Fortelný and Jůza [48]:(54)$$P_{c}\left(R\right)=\langle \begin{array}{c} P_{\mathrm{sph}}\\ P_{\mathrm{JM},\mathrm{Arb}} \end{array}\;\;\begin{array}{c} \mathrm{for}\;R\leq R_{L,\mathrm{FJ}}\\ \mathrm{for}\;R>R_{L,\mathrm{FJ}} \end{array}.$$

*R*_L,FJ_ is solution of the following equation:
*P*_sph_ = *P*_JM,Arb_(*R*_L,FJ_).(55)

*P*_sph_ is given by Equation (53) for the shear flow. Fitting of the results for the probability of coalescence of spherical particles in the extensional flow [73] leads to the following approximate expression [48]:(56)$$P_{sph,e}={\left[2.17781p^{0.69604}+0.77644\right]}^{-1}$$

All the above theories were derived assuming that the droplets are spherical till collision, and only binary collisions, not affected by the presence of other droplets, are important. Moreover, the results for Newtonian droplets in a Newtonian matrix can only be considered as conclusive. Applicability of Equation (47) to description of the matrix drainage and of the droplet pair rotation for general rheological model of viscoelastic matrix should be verified. To the best of our knowledge, the effect of the droplet elasticity on the coalescence has not been studied so far. The effect of the droplet anisometry in flow on *P*_c_ was studied by Patlazhan and Lindt [99] for systems with a small *p*. They used Janssen’s approach (Equation (42)) to calculate ellipsoidal droplets oriented in the flow direction. In calculation of *t*_c_ for the mobile interface model, they replaced radii of colliding droplets with their local radii of curvature at the point of contact in calculation of *R*_eq_. In calculation of the interaction time *t*_i_, they multiplied *t*_i_ = $\dot{\gamma }$^–1^ for spherical droplets by the ratio of distance between the short axis and contact point of ellipsoids to the distance between centers of ellipsoids in the direction perpendicular to flow. They found that *P*_c_ for elliptical droplets was higher than that for the spherical droplets of the same volume. In addition to the general limits of Janssen’s theory discussed above, the plausibility of Patlazhan’s and Lindt’s modification should also be an object of further study. To the best of our knowledge, the effect of other droplets on the coalescence of individual droplet pairs in a system with a high content of the dispersed phase has not been reliably addressed so far.

## 4. Competition between the Droplet Breakup and Coalescence

Tokita [100] considered average droplet size in steady shear flow. He assumed that the droplets were still monodisperse in size. In this case, Equation (1) reduces to:(57)$$Fn=\frac{4}{\pi }\dot{\gamma }P_{c}\phi n,$$
where *n* is number of spherical droplets in a volume unit and *ϕ* is volume fraction of the dispersed phase. Tokita assumed *P*_c_ independent of *R*; this is valid for conditions under which *P*_c_ = *P*_Sph_. The following equation was derived for breakup frequency using the assumption that total breakup energy consists of volume and interfacial energy of the droplet:(58)$$F=\frac{\eta_{ap}\dot{\gamma }}{E_{DK}+3\sigma /R},$$
where *η*_ap_ is the apparent viscosity of the blend and *E*_DK_ is the volume energy. Substitution from Equation (58) into Equation (57) and their solution leads to the following dependence of *R* on system parameters:(59)$$R=\frac{12\sigma P_{c}\phi }{\pi \eta_{ap}\dot{\gamma }-4P_{c}E_{DK}\phi }.$$

Equation (59) predicts reasonable shape of the dependence of *R* on *ϕ*, but it leads to *R* = 0 in the limit of *ϕ* → 0. This is in contradiction with generally accepted theories of the droplet breakup.

Steady droplet size and monodisperse system in shear flow were considered also by Fortelný and Kovář [101]. They assumed that *F* was a positive function of *Ca* for *Ca* > *Ca*_c_ which can be expanded into Taylor series on the right of *Ca*_c_:(60)$$F(Ca)={\left(\frac{\partial^{+}F}{\partial Ca^{+}}\right)}_{Ca_{c}}(Ca-Ca_{c})+.\frac{1}{2}{\left(\frac{\partial^{+2}F}{\partial Ca^{+2}}\right)}_{Ca_{c}}{\left(Ca-Ca_{c}\right)}^{2}+\ldots$$

This approach is general but it should be mentioned that derivatives in Equation (60) diverge if $F(Ca)\propto {\left(Ca-Ca_{c}\right)}^{a}$ with *a* < 1. Substitution of Equation (60) into Equation (57) leads to the linear dependence of *R* on *ϕ* if only the first term on the right of Equation (60) and *P*_c_ independent of *R* are considered:(61)$$R=R_{c}+\frac{4\sigma P_{c}}{\pi \eta_{m}f_{F}}\phi ,$$
where *f*_F_ is a function of the rheological properties of the blend components, independent of *R* and *ϕ*. Equation (61) reliably describes dependence of *R* on *ϕ* for some blends. Other blends show steeper than linear growth of *R* with *ϕ*.

Elmendorp and Van der Vegt [46,70] proposed estimation of the range of possible droplet sizes in steady shear flow. The smallest size of droplets that can burst is *R*_c_, which can be determined using the Taylor theory or empirical Equation (5) for Newtonian systems. The upper limit of the size of coalescing droplets was determined from the condition that *P*_c_ decreases to a negligible value in their model of coalescence. These conditions correspond to straight lines in the plot ln*R* vs. ln(*η*_m_$\dot{\gamma }$/*σ*), dividing it into four regions. Steady droplet radii lie either in the region relating to the dynamic equilibrium between the droplet breakup and coalescence is established or in the region where both breakup and coalescence are absent. From the regions where only breakup or only coalescence is operating, the droplet radii must pass to any of the former regions. The same approach was applied by Janssen [7,43,64], who calculated the maximum droplet size using his theory of coalescence.

Lyngaae–Jørgensen and Valenza [102] described a system where highly elongated drops burst into a large number of small droplets. This model assumes that dynamic equilibrium between small spherical and large ellipsoidal droplets is established in the steady state. A set of three equations for the volume fraction of large particles and length of their long and short half-axis was derived.

Huneault et al. [103] developed a computational model for droplet size evolution during mixing in a screw extruder. They considered that the blend components showed power law relations between shear stress and shear rate. They assumed that the droplet deformation took place only within the pressurized screw zones. Breakup into two fragments and coalescence of the droplets were considered for *Ca*_c_ < *Ca* < 4*Ca*_c_. The droplet fibrillation and disintegration was assumed for *Ca* > 4*Ca*_c_. The fiber breakup was considered either when blends entered a partially filled screw region or when the fiber diameter decreased below 1 μm. Parameter *C*_H_, characterizing the coalescence in a blend, was determined from the dependence of *R* on *ϕ* obtained during steady mixing in a batch mixer. The authors proposed the following equation for *R*:(62)$$R=R_{0}+{\left(1.5C_{H}Ca_{c}t_{B}^{*}\phi^{8/3}\right)}^{1/2},$$
where *R*_0_ is the droplet radius for *ϕ* = 0 and *t*_B_* is the dimensionless breakup time which is a function of *p* and is independent of *Ca*. The equation was derived by somewhat inconsistent procedure assuming breakup frequency independent of *R* and using Utracki’s theory of the coalescence [104], which is not consistent with commonly accepted theories of the shear-flow coalescence described in the preceding section.

Equation (1) was solved in [105] for steady shear flow using the assumption that a droplet bursts into two halves or with the same probability into two fragments having any volume. It was assumed that *F* was proportional to (*Ca* − *Ca*_c_)*^n^*, where n is a positive number, and that *P*_c_ decreases with a power of the average droplet radius. Scaling rules derived assuming that functions *C* and *F* are homogeneous, were used in solution of Equation (1). Algebraic equations were derived for the steady average droplet radius and for the characteristic time needed for the system transition to the steady state.

Janssen and Meijer [43] studied the evolution of droplet size in an extruder using a two-zone model. They assumed affine stretching of droplet and thread breakup in flow in the “strong” zone and thread breakup at rest and coalescence in the “weak” zone. The strong zone was modeled by extensional flow. The disturbance amplitude in the stretch thread described by Equation (20) was considered. Simple shear flow with a low $\dot{\gamma }$ was assumed in the weak zone. Janssen’s theory of coalescence for droplets with partially mobile interface was used. The model of a cascade of ideal mixers was used for the residence time distribution in the weak zone.

Patlazhan and Lindt [99] solved Equation (1) using expression for droplet breakup constructed by a combination of the results of Tomotika’s theory with the results for deformation of droplets in a system with low *p*. Gaussian distribution of volumes of daughter droplets was assumed. Coalescence was described by the modification of Janssen’s theory for ellipsoidal droplets mentioned in the preceding section. Droplet size distribution function as a function of the initial droplet size distribution, of *p* and of the average number of daughter droplets were calculated numerically.

Delamare and Vergnes [106] studied evolution of the droplet size distribution in a twin-screw extruder. They modeled the droplet breakup mechanism similarly to Hunealt et al. [103]: only two *Ca*_c_ instead of four *Ca*_c_ were considered as the boundary between regions of *Ca* in which breakup into two fragments or droplet fibrillation proceeds. They assumed breakup time, *t*_B_, increasing with *R* for *Ca*_c_ < *Ca* < 2 *Ca*_c_ and given by the Tomotika theory for fibers formed for *Ca* > 2 *Ca*_c_. Coalescence was described in the same way as in [43]. Average droplet diameters and local distribution of the droplet sizes was calculated numerically as functions of parameters of blends and of the extrusion process.

Milner and Xi [107] considered a batch mixer containing two regions: high-shear having small volume and low-shear with large volume. Ratio of residence times of a droplet in the high-shear and low-shear regions equal to the ratio of their volumes was assumed. One breakup of a droplet with *R* > *R*_c_ into two halves during its pass through the high shear region was considered. Only shear-flow-induced coalescence was assumed in the low-shear region. Probability of coalescence *P*_c_ was calculated by the Wang et al. theory [59] for spherical droplets. Proposed theory provides steady droplet size distribution by a numerical solution of derived equations.

Lyu et al. [108] assumed breakup frequency independent of *R* for *R*/*R*_c_ > 1 in their study of the effect of used theory of coalescence (Smoluchowski’s with *P*_c_ = 1, Wang’s et al. [59] for spherical droplets and Janssen’s for partially mobile interface) on agreement between calculated and experimentally determined droplet size evolution in steady shear flow. They found that none of the used coalescence theories led to satisfactory agreement with the experimental data.

Potente and Bastian [109] derived algorithm for calculation of the droplet size evolution during extrusion using the finite and boundary element methods for determination of the stress acts on the droplets during their trajectories. They characterized the flow field in the screw elements by the ratio, *λ*_r_, of the magnitude of the strain rate tensor to the sum of magnitudes of the strain rate and vorticity tensors. *Ca*_c_ was calculated by the equation with parameters determined from Grace’s experimental results [14]. Effective interfacial tension, given by Equation (5), stepwise breakup mechanism, and the breakup time increasing with *R*^0.37^ for *Ca* > *Ca*_c_ were considered for the description of the droplet breakup. Coalescence was calculated using Janssen’s method [43].

Fortelný [110] tried to evaluate plausibility of various expressions for the breakup time *t*_B_ (breakup frequency *F*) used in previous theories by graphic solution of Equation (57) with respect to *R*. A schematic graphic solution of Equation (57), used for this discussion, is shown in Figure 6. The dependence of *P*_c_ on *R* following from theories of Rother and Davis [60] and Fortelný and Jůza [71,73], i.e., *P*_c_ independent of *R* in the region of small *R* and its steep decrease after achieving certain limit value of *R*, was considered. It was shown that the form of the curve *F*(*R*) had a fundamental effect on the dependence of steady *R* on *ϕ*. Any function *F*(*R*) which does not meet the condition *F*(*R*_c_) = 0, i.e., *t*_B_(*R*_c_) → ∞, obviously leads to the steady *R* independent of *ϕ* for a certain region of *ϕ*. This type of the dependence has not been obtained experimentally so far. For functions *F*(*R*) with maximum at *R* > *R*_c_, Equation (57) has two solutions for a certain set of parameters.

Peters et al. [20] derived constitutive equation for liquid mixture based on the Lee and Park model of immiscible polymer blends [111]. The constitutive equation contains parameters related to the blend structure. Stepwise breakup of small droplets, stretching large droplets into filaments and their static and dynamic breakup and droplet coalescence were considered. A scheme for numerical calculation of the morphology evolution, considering the above events, was proposed. Breakup time was empirically determined by Grace [14] for the stepwise breakup, Janssen’s and Meijer’s procedure [7,43] for the description of the filament breakup, and Janssen’s and Meijer’s theory of coalescence [7,43] for blends with partially mobile interface were used in calculations. Results of the theory were compared with experimentally determined time dependence of rheological functions in various flow regimes, not with droplet size distribution.

Fortelný and Jůza [48] have recently formulated equations for calculation of steady droplet size in flowing immiscible polymer blends. They focused on calculation of the average size using assumption that the droplets were monodisperse. An approximate equation (Equation (51) or (54)), reflecting the dependence of *P*_c_ on system parameters resulted from theories of Rother and Davis [60] and Fortelný and Jůza [71,72,73], and was used for the description of the coalescence. Breakup frequency in the steady shear flow for *Ca*, not much higher than *Ca*_c_ was constructed from the experimental results of Cristini et al. [47] for the breakup time *t*_B_ and the number of fragments *n*_f_. The equation:(63)$$F=(n_{f}-1)t_{B}^{-1}$$
was used for a monodisperse system. Solution of Equation (57) with *F* given by Equation (63) with *t*_B_ expressed using Equation (15b) led to the equation:(64)$$a_{m}{\left(R*-1\right)}^{a_{c}+1/2}+{\left(R*-1\right)}^{1/2}-\frac{4(1+p)}{\pi }k_{1}P_{c}(R)\phi (R*-1)-\frac{4(1+p)}{\pi }k_{0}P_{c}(R)\phi =0$$
where *R** is the ratio of *R* and its critical value for breakup, *R*_c_ and *a*_m_ = 8.759 and *a*_c_ = 1.748, *k*_0_ = 4.3 and *k*_1_ = 27.7 are numerical constants determined from experimental data of Cristini et al. [47]. Unfortunately, data for the dependence of *n*_f_ on *R* − *R*_c_ show variance, which negatively affects the reliability of the derived dependence of the average *R* on *ϕ*. On the other hand, solution of Equation (64) is not much sensitive to values of *a*_m_ and *a*_c_. Solution of Equation (64) with *P*_c_(*R*) = *P*_Sph_ showed almost linear growth of *R** with *ϕ*. The rate of growth increased with *p*. Dependence of *R** on *ϕ* predicted by Equation (64) fairly matched that experimentally determined for polypropylene/ethylenepropylene elastomer and poly(lactic acid)/polycaprolactone blends for *ϕ* until about 0.2 (see Figure 7). However, critical radii for the droplet breakup, *R*_c_, and for steep decrease in *P*_c_, *R*_L_, determined experimentally [112,113] are much large than those predicted by Equations (3) and (55), respectively.

For *Ca* >> *Ca*_c_, the breakup frequency was constructed from *t*_B_ calculated as a sum of times of the droplet deformation into fibril and of the fibril breakup. This approach led to the equation [48]:(65)$$R{*}^{1/3}=\frac{4}{\pi }\frac{\phi }{g(p)Ca_{c}^{1/3}}P_{c}(R),$$
where *g*(*p*) is given by the equation:(66)$$g(p)=\frac{2^{5/2}}{3\pi \left[2+{\left(2\sqrt{2}\right)}^{\mu }\right]}x_{m}(p),$$
where μ = 0.65 and *x*_m_(*p*) is dominant wave number according to fibril breakup theory. *R** proportional to *ϕ*^3^ follows from Equation (65) for *P*_c_ = *P*_Sph_. However, dependence of *P*_c_ on average *R*, characteristic for a highly polydisperse system, should be considered in this case. Therefore, slower increase of *R** with *ϕ* should be characteristic for polydisperse systems. Generally, the dependence of the average *R* on *ϕ* can be different in intervals of small and large *ϕ*. Most dependences of average *R* on *ϕ*, determined experimentally at constant mixing conditions, can be expressed by quadratic form in *ϕ*. These dependences can be quite modeled well by the combination of the solutions of Equations (64) and (65).

The equation for the average droplet radius in steady extensional flow was derived [48] using breakup frequency following from Cox’s theory [12]. This equation did not provide reasonable dependence of *R* on *ϕ*, apparently because Cox’s theory is not plausible for *Ca* higher than *Ca*_c_.

## 5. Discussion of Problems with Prediction of the Droplet Size Formed by Steady Mixing

Many blends of synthetic polymers have been already successfully commercialized. It is probable a reason why the study of the relations between droplet size on the one hand and blend composition, parameters of the blend components and flow characteristics on the other hand have not been intensively studied during recent years. Sometimes theories used for the interpretation of experimental data have been chosen occasionally without consideration of their plausibility for a certain system, which can lead to false conclusions. We believe that the practical importance of knowledge of the above mentioned relations will be strongly enhanced in the near future due to attempts to commercialize blends of bio-polymers. The main features of correct description of the droplet breakup and coalescence in flowing polymer blends are summarized in the paragraph below. Other parts of this section deal with discussion of applicability of theoretical results obtained for blends of Newtonian liquids in simple flows to blends of viscoelastic liquids in complex flow fields.

The above results demonstrate that the dependence of the droplet breakup frequency on its radius is fundamental for the dependence of the average droplet radius on volume fraction of the dispersed phase. Dependence of the breakup frequency on system parameters needs further investigation. Different breakup mechanisms should be considered for blends with average *R* only slightly larger than *R*_c_ and with average *R* substantially larger than *R*_c_. *P*_c_ should be considered as a decreasing function of *p* in description of flow-induced coalescence in systems characterized with small *Ca*. Different dependences of *P*_c_ on the average droplet radius are valid for systems with narrow and broad droplet size distributions. These statements should be considered in derivation of any reliable theory of the phase structure evolution in polymer blends.

Most of the above theories, describing the droplet breakup, coalescence, and the competition between them, were derived for Newtonian systems in simple flows. Their results have been frequently used for prediction of the morphology of polymer blends formed during their mixing in batch mixers or extruders. However, polymers are viscoelastic substances showing shear thinning and flow fields in batch mixers and extruders are complex. The effect of the droplet and matrix elasticity on the droplet breakup has been studied rather broadly; main results are mentioned in Section 2. On the other hand, these results are not sufficient for construction of the dependence of breakup frequency on the droplet radius, viscosities, and elasticity parameters of the droplet and matrix and flow characteristics. The results of recent approximate studies of the effect of matrix elasticity on the probability of coalescence, *P*_c_, are summarized in Section 3. To our best knowledge, the effect of the droplet elasticity on *P*_c_ has not been addressed so far. Therefore, the effect of the matrix and droplet elasticity on coalescence needs further intensive investigation.

Components of polymer blends are mostly shear-thinning liquids. This must be taken into account when describing the droplet breakup and coalescence. Since the stress is continuous at the interface, viscosity and elasticity parameters of the matrix and droplets at constant stress related to the flowing blend should be considered. In contrast to other approaches [43,64], we believe that it should be applied also to the description of flow-induced coalescence because rheological properties are controlled by conformation of polymer molecules (deformation, entanglements) in flow.

Further aspect, not properly addressed so far, is the effect of surrounding droplets on the breakup of a droplet and coalescence of a certain droplet pair. The concept of substitution of *η*_m_ with the blend viscosity in calculation of *Ca* [42,43] should be verified for determination of *R*_c_ and its applicability to the description of the breakup and coalescence frequencies should be studied. It should be taken into account that, due to the slip at the interface, viscosity of a blend is frequently not equal to the viscosity calculated for the related emulsion using stick condition at the interface.

The above mentioned problems are common for the description of the phase structure evolution in simple flows, e.g., in rheometers, and in complex flow fields in mixing and processing devices. Specific problem of the description of the droplet size evolution in mixing and processing devices is proper modeling of the flow field in these devices. Substitution of complex, position-dependent, flow fields in mixing and processing devices with much simpler flows is unavoidable in a quantitative or semi-quantitative description of the particle size distribution during mixing and processing of polymer blends. So far, only a very limited care has been paid to this problem. Shear flow with the effective shear rate has been considered in evaluating the morphology formed in batch mixers [1,3,4]. The two-zone model, having zones with weak shear and strong extensional flows, was considered in modeling the flow field in a screw extruder [43,64].

Recently, we have compared experimentally determined average droplet sizes in polypropylene/ethylene-propylene rubber [112] and poly(lactic acid)/polycaprolactone [113] blends mixed in a batch mixer with the droplet sizes calculated using effective shear flow in the mixer. The effective shear rate, $\dot{\gamma }$_eff_, in the batch mixer was determined by the method [114] successfully applied to determine the flow curve from the dependence of the torque on the rotor speed of a batch mixer. For both blends, experimentally determined *R*_c_ were substantially larger than *R*_c_ calculated from Equation (3) using $\dot{\gamma }$_eff_. Agreement between calculated and measured value of *R*_c_ was not improved using *σ*_eff_, given by Equation (5), instead of equilibrium *σ* in calculation of *Ca*. Using of $\dot{\gamma }$_eff_ leads to the prediction that the collision efficiency, *P*_c_, is negligibly small for both blends. It was in strong contradiction with experimentally detected pronounced growth of the average droplet radius with volume fraction of the dispersed phase. These results indicate that improvement in modeling of flow fields in mixing and processing devices and in description of breakup and coalescence in viscoelastic systems is a necessary condition for at least semi-quantitative prediction of the droplet size.

## Figures and Tables

**Figure 1 polymers-11-00761-f001:**
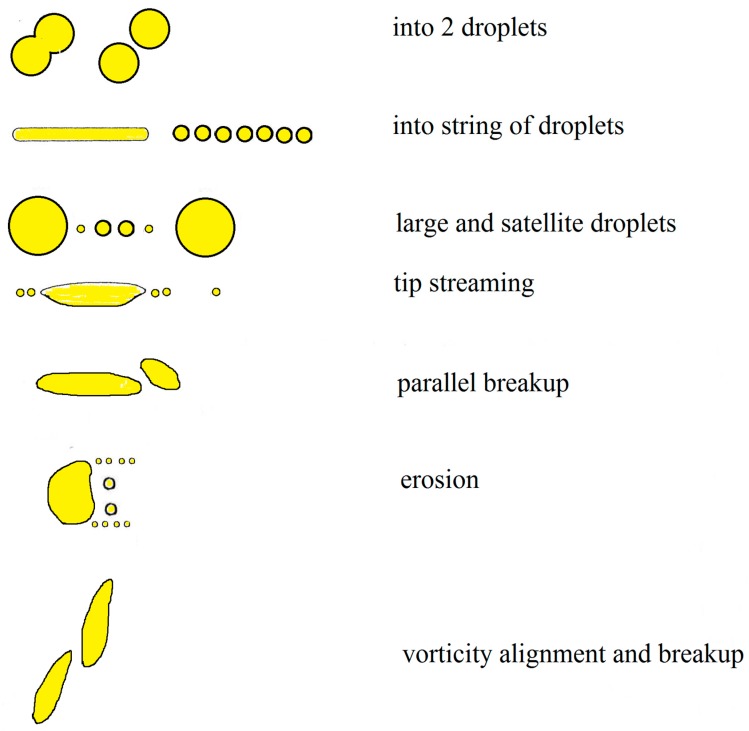
Schemes of various breakup mechanisms.

**Figure 2 polymers-11-00761-f002:**
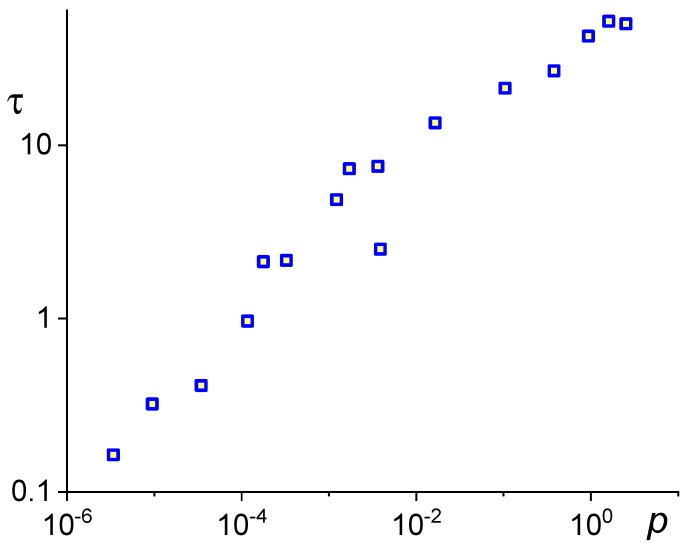
Reduced breakup time *τ* = *t*_B_*σ*/(*R*_c_*η*_m_) determined by Grace [14] for different values of viscosity ratio *p*.

**Figure 3 polymers-11-00761-f003:**
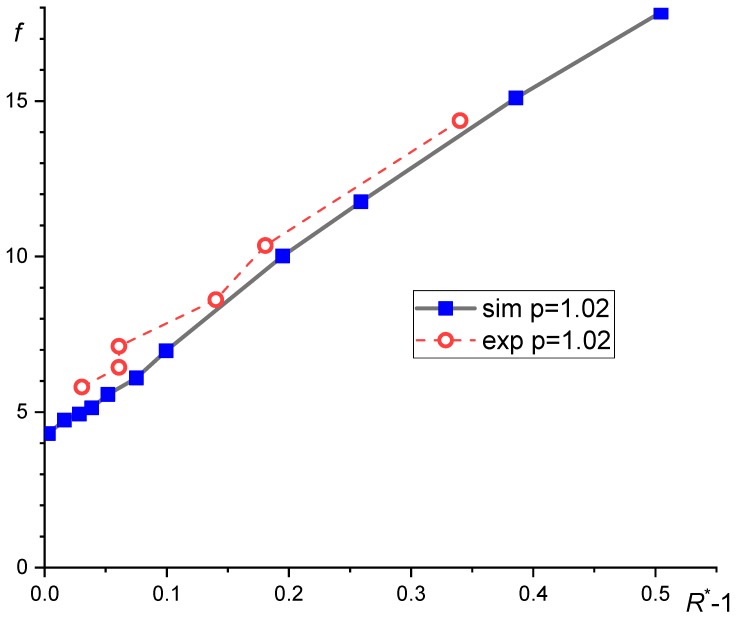
Dependence of scaled time function *f* = $\dot{\gamma }$*t*_B_ (*R** − 1)^1/2^ on reduced droplet radius *R** found by Cristini [47] using simulation (solid line, full squares) and experiment (dashed line, empty circles).

**Figure 4 polymers-11-00761-f004:**
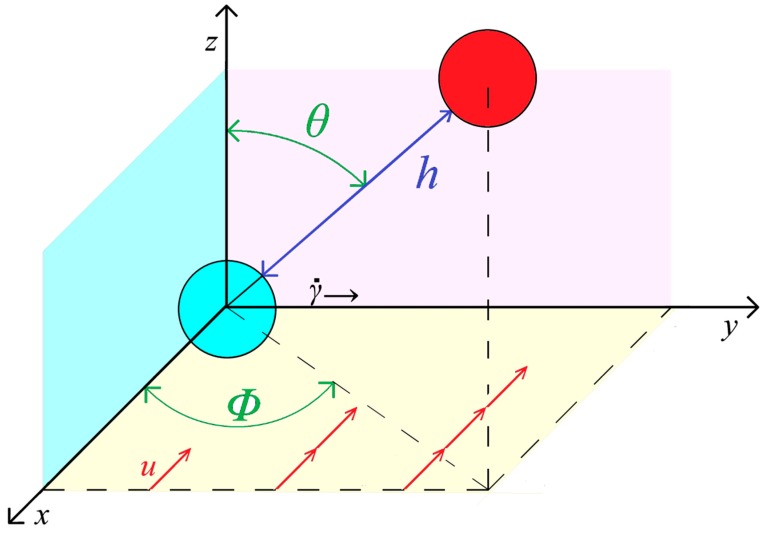
System geometry and parameters for shear-flow-induced coalescence described here.

**Figure 5 polymers-11-00761-f005:**
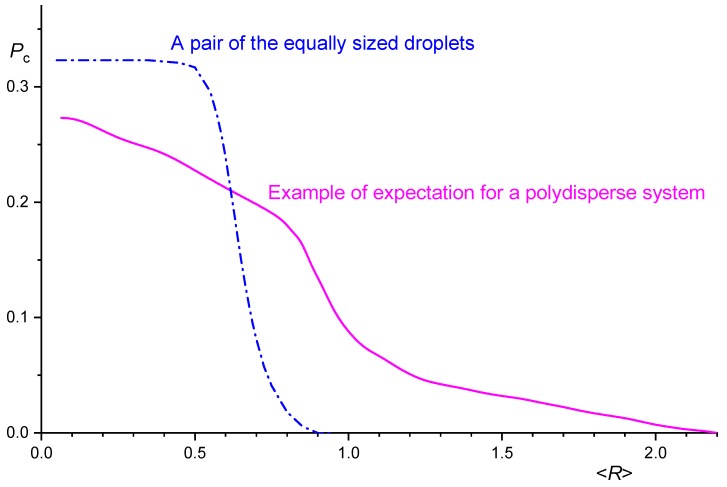
Dependence of collision efficiency (*P*_c_) on average effective droplet radius for monodisperse (dash-dotted) and polydisperse (solid curve) systems.

**Figure 6 polymers-11-00761-f006:**
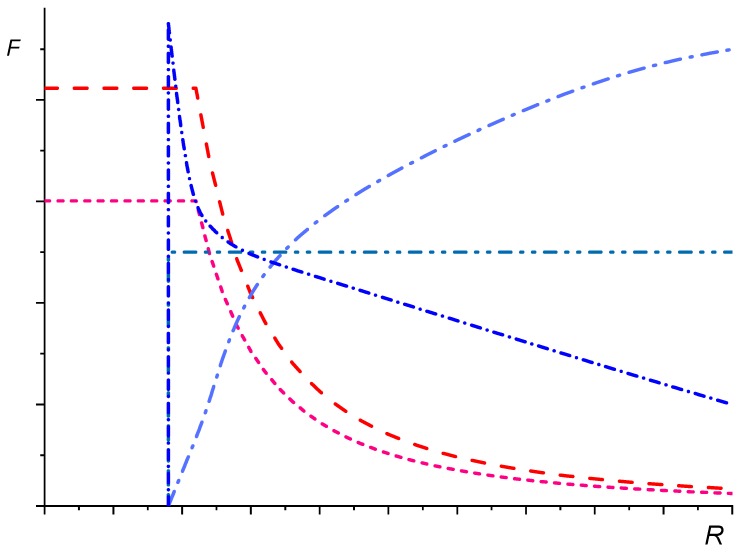
Equilibrium breakup (dash-dotted curves)—coalescence (dashed curves) as function of breakup or fusion frequency on droplets radius. The figure is based on analysis in [110].

**Figure 7 polymers-11-00761-f007:**
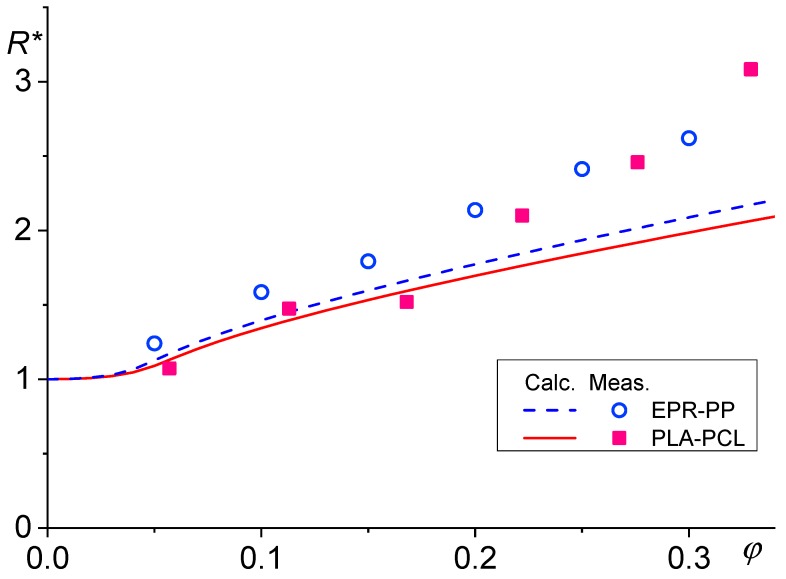
Reduced droplet sizes calculated using Equation (64) and experimental sizes for blends ethylene-propylene rubber (EPR)-polypropylene (PP) [112] (dashed line and empty circles) and poly(lactic acid) (PLA)-poly(caprolactone) (PCL) [113] (solid line, full squares).

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
