# Peer review of "Description of the Droplet Size Evolution in Flowing Immiscible Polymer Blends"

_polymers, 2019, doi:10.3390/polym11050761_

Round 1
Reviewer 1 Report
This review is nicely organized providing a quite comprehensive theoretical overview of immiscible polymer blends behaviour. Droplet formation and behaviour during processing and melt flow are important factors of applicable, industrial polymers, that older and of course younger polymer researchers should be aware of and also find easily in a gathered consise article that this herein review.
However, the phrase in the last paragraph of the introduction part that "current state of the art in the field" is presented, is an overstatement since none of the included references are of the recent few years. Perhaps the authors could include a few more recent results from current literature, if any, that could make the review more appealing to younger researchers and the journal's audience. If this is not possible then the particular phrase should be corrected.
Moreover, some actual paradigms could be included from applicable, industrial polymer blends in order to facilitate the readers and help them correlate the theory to actual practises.
Aside these, a small corrections and possible inclusions, the manuscript is publishable in its current form.
Author Response
This review is nicely organized providing a quite comprehensive theoretical overview of immiscible polymer blends behaviour. Droplet formation and behaviour during processing and melt flow are important factors of applicable, industrial polymers, that older and of course younger polymer researchers should be aware of and also find easily in a gathered consise article that this herein review.
R1:1: However, the phrase in the last paragraph of the introduction part that "current state of the art in the field" is presented, is an overstatement since none of the included references are of the recent few years. Perhaps the authors could include a few more recent results from current literature, if any, that could make the review more appealing to younger researchers and the journal's audience. If this is not possible then the particular phrase should be corrected.
Answer: Formulation of the last paragraph in Introduction has been changed in the revised manuscript. New formulation states that the aim of the paper is to summarize results important for prediction of the droplet size in flowing polymer blends. In last few years, papers focused to various aspects of the droplet breakup and coalescence have been published. Utilization of results (mostly numerical) of these papers for prediction of the droplet size in flowing blends is difficult. We did not find any attempt to do it in the literature. Therefore, we decided not extent the review by description of results of these papers.
R1:2: Moreover, some actual paradigms could be included from applicable, industrial polymer blends in order to facilitate the readers and help them correlate the theory to actual practises.
Answer: New paragraph briefly discussing present methods of evaluation of experimental data obtained at mixing and processing of polymer blends has been added to the first part of section 5. We are ready to make further modification in case of more specific suggestions.
R1:3:Aside these, a small corrections and possible inclusions, the manuscript is publishable in its current form.
Answer: Corrections made.
Reviewer 2 Report
The authors present a review on the theoretical progress of descriptions of the evolution of structure of binary polymer blends in simplified flow regimes. The relevance and limitations of theoretical approaches to simple and well established polymer processing methods is established.
There is some mention of the molecular mechanisms of interactions between droplets on page 14 (line 416) but I feel this could be discussed a little earlier in the text.
There are terms in the text which could be related to Figure 1 and clarified: e.g. fibrillation, daughter droplets.
On the whole the presentation is logical, well-structured and clear but hampered by some minor presentation issues, particularly with respect to grammar. There are numerous corrections necessary and in particular there are many missing articles.
Specific comments
Page 1
Line 36 remove “their”
Line 37: “An important…”
Page 3
Line 100: “daughter’s droplets” should be “daughter droplets” - the error is common in the manuscript.
Line 113: “is given by dynamic” should be “is given by the dynamic…”
Page 5
Figure 1 is not explicitly referenced in the manuscript text. There are a number of places where the reference should be quite useful.
Line 170: “Further parameter” should be “A further parameter…”
Page 8
Line 205: “daughter’s droplets” should be “daughter droplets”
Line 208: “daughter’s droplets” should be “daughter droplets”
Page 9
Line 232: insert “the” between “that” and “breakup”
Line 246: insert “an” between “of” and “elongated”
Page 10
line 270: “Number of daughter’s droplets” should be “The number of daughter droplets”
Line 276: “for a number daughter’s droplets” should be “for the number of daughter droplets”
Line 278 “A further possible”
Line 293: “daughter’s droplets” should be “daughter droplets”
Page 11
Line 319: “of a pair of droplets”?
Page 12
Line 332: “The hydrodynamic interaction, leading to a decrease in Pc, prevails for them.”
Line 338: “even if the magnitude”
Line 347: “not flatten” should be “not flattened”
Line 354: “The equation”
Page 16
Figure 5 is not referenced explicitly in the text. I also feel that more rigor could be applied to the definition of “polydispersity” but this is a very small point.
Page 21
Figure 6 is not explicitly referred to in the text.
Author Response
Reviewer 2
R2:1: The authors present a review on the theoretical progress of descriptions of the evolution of structure of binary polymer blends in simplified flow regimes. The relevance and limitations of theoretical approaches to simple and well established polymer processing methods is established.
R2:2: There is some mention of the molecular mechanisms of interactions between droplets on page 14 (line 416) but I feel this could be discussed a little earlier in the text.
Answer: A brief discussion of molecular forces has been added on page 12 (below Eq. (28)).
R2:3: There are terms in the text which could be related to Figure 1 and clarified: e.g. fibrillation, daughter droplets.
Answer: Terms relating to various mechanisms of the droplet breakup have been clarified and related to Fig. 1 in the revised text.
R2:4: On the whole the presentation is logical, well-structured and clear but hampered by some minor presentation issues, particularly with respect to grammar. There are numerous corrections necessary and in particular there are many missing articles.
Answer: All suggested corrections of grammar have been accepted in the revised text. English of the paper was once more checked.
R2:5: Line 36 remove “their”
Answer: Corrected.
R2:6: Line 37: “An important…”
Answer: Corrected.
R2:7: Line 100: “daughter’s droplets” should be “daughter droplets” - the error is common in the manuscript.
Answer: Corrected.
R2:8: Line 113: “is given by dynamic” should be “is given by the dynamic…”
Answer: Corrected.
R2:9:Figure 1 is not explicitly referenced in the manuscript text. There are a number of places where the reference should be quite useful.
Answer: Several references have been added where appropriate. Figure 1 is explicitly referenced several times in the revised manuscript.
R2:10: Line 170: “Further parameter” should be “A further parameter…”
Answer: Corrected.
R2:11: Line 205: “daughter’s droplets” should be “daughter droplets”
Answer: Corrected.
R2:12: Line 208: “daughter’s droplets” should be “daughter droplets”
Answer: Corrected.
R2:13: Line 232: insert “the” between “that” and “breakup”
Answer: Corrected.
R2:14: Line 246: insert “an” between “of” and “elongated”
Answer: Corrected.
R2:15: Line 270: “Number of daughter’s droplets” should be “The number of daughter droplets”
Answer: Corrected.
R2:16: Line 276: “for a number daughter’s droplets” should be “for the number of daughter droplets”
Answer: Corrected.
R2:17: Line 278 “A further possible”
Answer: Corrected.
R2:18: Line 293: “daughter’s droplets” should be “daughter droplets”
Answer: Corrected.
R2:19: Line 319: “of a pair of droplets”?
Answer: Corrected.
R2:20: Line 332: “The hydrodynamic interaction, leading to a decrease in Pc, prevails for them.”
Answer: Corrected.
R2:21: Line 338: “even if the magnitude”
Answer: Corrected.
R2:22: Line 347: “not flatten” should be “not flattened”
Answer: Corrected.
R2:23: Line 354: “The equation”
Answer: Corrected.
R2:24:Figure 5 is not referenced explicitly in the text.
Answer: Sentence referencing Figure 5 has been added to the text.
R2:25: I also feel that more rigor could be applied to the definition of “polydispersity” but this is a very small point.
Answer: Sentence specifying polydispersity has been added to the text.
R2:26:Figure 6 is not explicitly referred to in the text.
Answer: A sentence referred Figure 6 has been added to the text.